# Analytic Solution and Noether Symmetries for the Hyperbolic Inflationary Model in the Jordan Frame

Andronikos Paliathanasis [1,2] 

1   Institute of Systems Science, Durban University of Technology, P.O. Box 1334, Durban 4000, South Africa; anpaliat@phys.uoa.gr
2   Instituto de Ciencias Físicas y Matemáticas, Universidad Austral de Chile, Valdivia 5090000, Chile

**Abstract:** The Noether symmetry analysis is applied for the study of a multifield cosmological model in a spatially flat FLRW background geometry. The gravitational Action Integral consists of two scalar fields, the Brans–Dicke field and a second scalar field minimally coupled to gravity. However, the two scalar fields interact in kinetic terms. This multifield has been found to describe the equivalent of hyperbolic inflation in the Jordan frame. The application of Noether's theorems constrains the free parameters of the model so that conservation laws exist. We find that the field equations form an integrable dynamical system, and the analytic solution is derived.

**Keywords:** cosmology; scalar field; inflation; analytic solutions; noether symmetries



## 1. Introduction

Scalar fields play an important role in the description of cosmological evolution [1]. With the introduction of scalar fields in the Einstein–Hilbert Action, the new degrees of freedom drive the dynamics of the cosmological parameters such that they explain the cosmological observations [2,3]. The quintessence scalar field model is a very simple model that describes the so-called dark energy and is responsible for the late-time acceleration phase of the universe [4,5]. On the one hand, inflation [6] has been proposed to solve the flatness, the horizon and the isotropization problems. Inflation describes a very rapid acceleration phase during the early stage of the universe, and it is attributed to the inflation field [7–9]. There are a plethora of proposed scalar field models in the literature; see, for instance [10–17], and references therein.

In the middle of the previous century, Brans and Dicke [18] proposed a gravitational model with a scalar field that satisfies Mach's principle. Indeed, the existence of the scalar field is essential for the physical space, and the scalar field interacts with the gravity in the Action Integral; that is, the scalar field is non-minimally coupled to gravity. Generalizations of the Brans–Dicke model are known as scalar-tensor theories [19]. In [20], Hordenski derived the most general Action Integral for the scalar-tensor theory. The Brans–Dicke theory is defined on the Jordan frame [21], while the gravitational model depends upon a free parameter known as the Brans–Dicke parameter. The Brans–Dicke model has been used as a model for the description of dark energy [22] and the inflationary epoch [23–25].

A two-scalar field model that has drawn the attention of cosmologists in recent years is the Chiral model [26–28]. In the Chiral model, the two scalar fields are minimally coupled to gravity; that is, they are defined in the Jordan frame. However, the two scalar field interact in the kinetic term. Specifically, from the kinetic components of the scalar fields, we can define a second-rank tensor, which for the Chiral model is a two-dimensional hyperbolic sphere. Thus, there is no "coordinate" system where there is no interaction between the two fields. This is in contrast to the quintom model in which the kinetic components define the two-dimensional flat space [29]. This specific two-scalar field model has been widely studied in the literature [30–33], and various extensions for which the scalar fields may have negative energy density have been proposed before [34,35].

For a specific potential function, the Chiral model provides a very interesting scaling solution in which the two scalar fields contribute to the cosmological fluid [36,37]. The scaling solution describes acceleration, and the solution is described as hyperbolic inflation or hyperinflation. Because of the existence of the second scalar field in the hyper-inflation, the curvature perturbations depend upon the number of e-folds [38], while the initial conditions at the start and at the end of the inflation can be different [38], and the non-Gaussianities in the power spectrum are supported by this model [39]. Recently, in [40], a multiscalar field model was proposed consisting of two-scalar fields, the Brans–Dicke field and a second field, which is coupled to the Brans–Dicke field in kinetic terms, but it is minimally coupled to gravity. The latter model is defined in the Jordan frame. However, under a conformal map, the equivalent Action Integral in the Einstein frame is that of the Chiral theory. The extended Brans–Dicke theory admits an asymptotic scaling solution, which has similar dynamical properties to the hyperbolic inflationary solution for the Chiral model. Indeed, the asymptotic solution describes inflation, in which the two scalar fields contribute to the cosmological solution, while this specific solution corresponds to a spiral attractor. This dynamical property remains invariant for the two models under the conformal transformation, which relates the two theories.

In this study, we investigate the conservation laws and the integrability properties of the hyperbolic inflationary model in the Jordan frame. For the purposes of this study, we make use of the property that the gravitational field equations admit a minisuperspace description so that the Noether symmetry analysis [41] can be applied. Noether's theorems provide a systematic approach for the determination of infinitesimal transformations, which leave the variational principle invariant. Moreover, the generators of the infinitesimal invariant transformations can be used in a simple way to construct conservation laws. Because of the simplicity of the applications of Noether's theorems and of the importance of the given results, Noether symmetries have been the subject of study in various gravitational systems [42–46]. The plan of the paper is as follows.

In Section 2, we present the cosmological model of our consideration, and we derive the minisuperspace and the point-like Lagrangian which describes the field equations. The basic properties and definitions for the theories of point transformations are given in Section 3. Moreover, we find the Noether symmetries, and we construct the corresponding conservation laws for the field equations. In Section 4, we determine the analytic solution for the cosmological model of our analysis. We define canonical variables and derive the analytic solution. Finally, in Section 5, we summarize our results.

## 2. Field Equations

The cosmological model of our consideration is that of a spatially flat Friedmann–Lemaître–Robertson–Walker (FLRW) geometry described by the line element

$$ds^2 = dt^2 - a^2(t)\left(dx^2 + dy^2 + dz^2\right), \tag{1}$$

where $a(t)$ is the scale factor.

The field equations follow from the variation of the Action Integral

$$S_A = \int dx^4 \sqrt{-g}\left[\frac{1}{2}\phi R + \frac{1}{2}\frac{\omega_{BD}}{\phi}g^{\mu\nu}\phi_{;\mu}\phi_{;\nu} + \frac{1}{2}F^2(\phi)g^{\mu\nu}\psi_{;\mu}\psi_{;\nu} + V(\phi)\right], \tag{2}$$

where $\phi(x^\kappa)$ is the Brans–Dicke field, $\omega_{BD}$ is the Brans–Dicke parameter, $V(\phi(x^\kappa))$ is the potential function, and $\psi(x^\kappa)$ is the second scalar field minimally coupled to gravity.

For the line element (1) and the Action Integral (2), we derive the field equations

$$-3H^2 = 3H\frac{\dot{\phi}}{\phi} - \frac{\omega_{BD}}{2}\left(\frac{\dot{\phi}}{\phi}\right)^2 - \frac{1}{2}\frac{F^2(\phi)}{\phi}\dot{\psi}^2 - \frac{1}{\phi}V(\phi), \tag{3}$$

$$- \left(3\phi H^2 + 2\phi \dot{H}\right) = 2H\dot{\phi} + \frac{\omega_{BD}}{2\phi}\dot{\phi}^2 + \frac{1}{2}F^2(\phi)\dot{\psi}^2 + \ddot{\phi} - V(\phi) \tag{4}$$

$$\omega_{BD}\left(\ddot{\phi} - \frac{1}{2}\left(\frac{\dot{\phi}}{\phi}\right)^2 + 3H\dot{\phi}\right) + 6H^2\phi + \phi\left(3\dot{H} + V_{,\phi} - \frac{1}{2}\left(F^2\right)_{,\phi}\dot{\psi}^2\right) = 0, \tag{5}$$

$$\ddot{\psi} + 3H\dot{\psi} + \left(\ln\left(F^2\right)\right)_{,\phi}\dot{\phi}\dot{\psi} = 0, \tag{6}$$

in which we have assumed that the scalar fields inherit the symmetries of the background space; that is, $\phi(x^\kappa) = \phi(t)$, $\psi(x^\kappa) = \psi(t)$, $H = \frac{\dot{a}}{a}$ is the Hubble function, and $\dot{a} = \frac{da}{dt}$.

It is easy to see that the cosmological field equations follow from the variation of the point-like Lagrangian function

$$\mathcal{L}(a, \dot{a}, \phi, \dot{\phi}, \psi, \dot{\psi}) = 3a\phi\dot{a}^2 + 3a^2\dot{a}\dot{\phi} - \frac{\omega_{BD}}{2\phi}a^3\dot{\phi}^2 - \frac{1}{2}a^3F^2(\phi)\dot{\psi}^2 + a^3V(\phi). \tag{7}$$

The constraint Equation (3) can be seen as a Hamiltonian constraint for the autonomous dynamical system.

From the point-like Lagrangian, we define the minisuperspace, which has three-dimensional line element

$$ds_\gamma^2 = 6a\phi da^2 + 6a^2 dad\phi - \frac{\omega_{BD}}{\phi}a^3 d\phi^2 - a^3 F^2(\phi)d\psi^2. \tag{8}$$

Hyperbolic inflation in the Jordan frame is recovered when $F(\phi) = F_0\phi^\kappa$ and $V(\phi) = V_0\phi^\lambda$. Hence, these two functions are considered in the following section.

### 3. Noether Symmetries and Conservation Laws

We review the basic definitions concerning invariant point transformations and Noether symmetries of systems of second-ordinary differential equations.

Assume the dynamical system

$$\ddot{\mathbf{y}} = \omega(t, \mathbf{y}, \dot{\mathbf{y}}). \tag{9}$$

Then, a vector field

$$X = \xi(t, \mathbf{y})\partial_t + \mathbf{J}(t, \mathbf{y})\partial_\mathbf{y} \tag{10}$$

in the augmented space $\{t, x^i\}$ is a point symmetry of the system of differential Equation (9) if the following condition is satisfied [46]

$$X^{[2]}(\ddot{\mathbf{y}} - \omega(t, \mathbf{y}, \dot{\mathbf{y}})) = 0, \tag{11}$$

where $X^{[2]}$ is the second prolongation of $X$ defined as follows

$$X^{[2]} = \xi\partial_t + \mathbf{J}\partial_\mathbf{y} + \left(\mathbf{j} - \dot{\mathbf{y}}\dot{\xi}\right)\partial_{\dot{\mathbf{y}}} + \left(\mathbf{j} - \dot{\mathbf{y}}\ddot{\xi} - 2\ddot{\mathbf{y}}\dot{\xi}\right)\partial_{\ddot{\mathbf{y}}}. \tag{12}$$

Thus, if $X$ is a symmetry vector, then under the infinitesimal transformation

$$t' = t + \varepsilon\xi(t, \mathbf{y}), \ y' = y + \varepsilon\mathbf{J}(t, \mathbf{y}), \tag{13}$$

the dynamical system (9) remains invariant, which means that trajectories of solutions lead to vector field $X$.

Condition (11) is equivalent to the relation

$$\left[X^{[1]}, A\right] = \lambda(x^a)A, \tag{14}$$

where $X^{[1]}$ is the first prolongation of $X$, and $A$ is the Hamiltonian vector field

$$A = \partial_t + \dot{\mathbf{y}}\partial_{\mathbf{y}} + \omega(t, \mathbf{y}, \dot{\mathbf{y}})\partial_{\mathbf{y}}. \tag{15}$$

If the system of differential equations results from a first-order Lagrangian $\mathcal{L} = \mathcal{L}(t, \mathbf{y}, \dot{\mathbf{y}})$, then a Lie symmetry $X$ of the system is a Noether symmetry of the Lagrangian if the additional condition

$$X^{[1]}\mathcal{L} + \mathcal{L}\frac{d\xi}{dt} = \frac{df}{dt} \tag{16}$$

is satisfied, where $f = f(t, \mathbf{y})$ is a boundary function and

$$X^{[1]} = \xi\partial_t + \mathbf{J}\partial_{\mathbf{y}} + (\mathbf{j} - \dot{\mathbf{y}}\dot{\xi})\partial_{\dot{\mathbf{y}}}. \tag{17}$$

According to Noether's second theory, to every symmetry, there corresponds a first integral (a Noether integral) of the system of Equations (9), which is given by the formula:

$$I(X) = \xi E_H - \frac{\partial\mathcal{L}}{\partial\dot{\mathbf{y}}}\mathbf{J} + f, \tag{18}$$

where $E_H(t, \mathbf{y}, \dot{\mathbf{y}})$ is the Hamiltonian function of $\mathcal{L}(t, \mathbf{y}, \dot{\mathbf{y}})$.

Consider now the infinitesimal transformation

$$t' = t + \varepsilon\xi(t, a, \phi, \psi) , \quad a' = a + \varepsilon\eta^a(t, a, \phi, \psi) , \tag{19}$$

$$\phi' = \phi + \varepsilon\eta^\phi(t, a, \phi, \psi) , \quad \psi' = \psi + \varepsilon\eta^\psi(t, a, \phi, \psi) , \tag{20}$$

and generation of the vector field

$$X = \xi\partial_t + \eta^a\partial_a + \eta^\phi\partial_\phi + \eta^\psi\partial_\psi. \tag{21}$$

Then, the application of Noether's condition (16) for the Lagrangian function (7) with $F(\phi) = F_0\phi^\kappa$ and $V(\phi) = V_0\phi^\lambda$ gives a system of differential equations, which constrain the infinitesimal parameters. The results are summarized in the following propositions.

**Proposition 1.** *The point-like Lagrangian (7) with $F(\phi) = F_0\phi^\kappa$ and $V(\phi) = V_0\phi^\lambda$ for arbitrary values of the parameters $\kappa, \lambda$ admits the Noether symmetries $X_1 = \partial_t$, $X_2 = \partial_\psi$. However, when $\kappa = \frac{1}{2}$, $\lambda = 1$, there exist the additional symmetry vectors $X_3 = a\partial_a - 3\phi\partial_\phi$, $X_4 = -\frac{a}{3}\psi\partial_a +$ $\phi\psi\partial_\phi + \frac{2}{F_0^2}\ln\left(\frac{a}{\phi^{1+\omega_{BD}}}\right)\partial_\psi$; while for $\lambda = 2\kappa$, $\kappa = \frac{\sqrt{3(3+2\omega_{BD})}}{4} + \frac{3}{4}$, the field equations admit the extra Noether symmetries $\bar{X}_3 = \phi^{\beta_1}a^{\beta_2}\left(\partial_a - \frac{6}{\sqrt{3(3+2\omega_{BD})}}\frac{\phi}{a}\partial_\phi\right)$ where $\beta_1 = \frac{\sqrt{3}(\omega_{BD}+1)}{2\sqrt{(3+2\omega_{BD})}} -$ $\frac{1}{2}$ , $\beta_2 = -\frac{1}{2} - \frac{3}{2\sqrt{3(3+2\omega_{BD})}}$ and $\bar{X}_4 = \phi^{\beta_1}a^{\beta_2}\left(-\psi\partial_\alpha + \frac{\psi\phi}{a}\partial_\phi + \frac{8(\omega_{BD}+\frac{3}{2})}{F_0^2(\sqrt{3(3+2\omega_{BD})}+1)a}\phi^{-\frac{1+\sqrt{3(3+2\omega_{BD})}}{2}}\partial_\psi\right)$. Moreover, for arbitrary value of $\kappa$ and $\lambda = 1$, there exist the Noether symmetry vector $X_5 = -\frac{a}{3}\partial_a + \phi\partial_\phi - \frac{\sqrt{3(3+2\omega_{BD})}+1}{4F_0^2}\psi\partial_\psi$.*

**Proposition 2.** *According to Noether' second theorem and for expression (18), the cosmological model of our consideration admits the conservation laws $I(X_1) = E_H$, $I(X_2) = a^3F_0^2\phi^{2\kappa}\dot{\psi}$ for arbitrary values of the free parameters $\kappa, \lambda$. For $(\kappa, \lambda) = \left(\frac{1}{2}, 1\right)$, the additional conservation laws are*

$$I(X_3) = a\left(6a\phi\dot{a} + 3a^2\dot{\phi}\right) - 3\phi\left(3a^2\dot{a} - \frac{\omega_{BD}}{\phi}a^3\dot{\phi}\right) , \tag{22}$$

$$I(X_4) = -\frac{a}{3}\psi\left(6a\phi\dot{a} + 3a^2\dot{\phi}\right) + \phi\psi\left(3a^2\dot{a} - \frac{\omega_{BD}}{\phi}a^3\dot{\phi}\right) + 2a^3\phi^{2\kappa}\dot{\psi}\ln\left(\frac{a}{\phi^{1+\omega_{BD}}}\right). \tag{23}$$

*For $(\kappa, \lambda) = (\kappa, 2\kappa)$, $\kappa = \frac{\sqrt{3(3+2\omega_{BD})}}{2} + \frac{3}{4}$, the corresponding conservation laws are*

$$I(\bar{X}_3) = \phi^{\beta_1} a^{\beta_2} \left( \left( 6a\phi\dot{a} + 3a^2\dot{\phi} \right) - \frac{6}{\sqrt{3(3 + 2\omega_{BD})}} \phi \left( 3a\dot{a} - \frac{\omega_{BD}}{\phi} a^2\dot{\phi} \right) \right), \qquad (24)$$

$$I(\bar{X}_4) = \phi^{\beta_1} a^{\beta_2} \left( -\psi \left( 6a\phi\dot{a} + 3a^2\dot{\phi} \right) + \psi\phi \left( 3a\dot{a} - \frac{\omega_{BD}}{\phi} a^2\dot{\phi} \right) + \frac{8\left( \omega_{BD} + \frac{3}{2} \right) a^2 \dot{\psi}}{\left( \sqrt{3(3 + 2\omega_{BD})} + 1 \right)} \phi^{2\kappa - \frac{1+\sqrt{3(3+2\omega_{BD})}}{2}} \right). \qquad (25)$$

*Finally, for arbitrary $\kappa$ and $\lambda = 1$, the additional conservation law is*

$$I(X_5) = -\frac{a}{3} \left( 6a\phi\dot{a} + 3a^2\dot{\phi} \right) + \phi \left( 3a^2\dot{a} - \frac{\omega_{BD}}{\phi} a^3\dot{\phi} \right) - \frac{\sqrt{3(3 + 2\omega_{BD})} + 1}{4} \phi^{2\kappa} \psi\dot{\psi}. \qquad (26)$$

We observe that the set of the conservation laws $(I(X_1), I(X_2), I(X_3))$ and $(I(X_1), I(X_2), I(\bar{X}_3))$ are independent and in involution, that is, $\{I(X_A), I(X_B)\} = 0$, $A, B = 1, 2, 3$ and $\{,\}$ is the Poisson bracket. Consequently, according to Liouville's theorem, the field equations of this two-dimensional system are integrable. Specifically, because they admit additional conservation laws, they are super-integrable [47]. For these two cases, we proceed with the derivation of the analytic solutions.

## 4. Analytic Solutions

The procedure that we apply for the derivation of the analytic solutions is summarized in the following steps. For the vector fields $X_A$, we find the normal variables by solving the system of differential equations

$$X_A(F(a, \phi, \psi)) = 0. \qquad (27)$$

We write the field equations in the new coordinates, and we solve the resulting system.

### 4.1. Model A

For the first case of our analysis $(\kappa, \lambda) = \left( \frac{1}{2}, 1 \right)$, and the symmetry vector $X_3$, we determine the normal coordinates $(a, \Phi, \psi)$ where

$$\phi = \frac{\Phi}{a^3}. \qquad (28)$$

Thus, the point-like Lagrangian (7) is

$$\mathcal{L}\left( a, \dot{a}, \Phi, \dot{\Phi}, \psi, \dot{\psi} \right) = \frac{3}{2}(4 + 3\omega_{BD})\Phi \left( \frac{\dot{a}}{a} \right)^2 - 3(1 + \omega_{BD}) \left( \frac{\dot{a}}{a} \right) \dot{\Phi} + \frac{\omega_{BD}}{2} \frac{\dot{\Phi}^2}{\Phi} + \frac{1}{2} F_0^2 \Phi\dot{\psi}^2 - V_0\Phi. \qquad (29)$$

Consequently, the field equations are

$$\frac{3}{2}(4 + 3\omega_{BD})\Phi \left( \frac{\dot{a}}{a} \right)^2 - 3(1 + \omega_{BD}) \left( \frac{\dot{a}}{a} \right) \dot{\Phi} + \frac{\omega_{BD}}{2} \frac{\dot{\Phi}^2}{\Phi} + \frac{1}{2} F_0^2 \Phi\dot{\psi}^2 + V_0\Phi = 0, \qquad (30)$$

$$(4 + 3\omega_{BD}) \left( \Phi\ddot{a} + \dot{a}\dot{\Phi} - \Phi\frac{\dot{a}^2}{a} \right) - (1 + \omega_{BD})a\ddot{\Phi} = 0, \qquad (31)$$

$$3(2 + \omega_{BD})\dot{a}^2 + 6(1 + \omega_{BD}a)\ddot{a} + a^2 \left( \omega_{BD} \left( \frac{\dot{\Phi}^2}{\Phi^2} - 2\frac{\ddot{\Phi}}{\Phi} \right) + \left( F_0^2\dot{\psi}^2 - 2V_0 \right) \right) = 0. \qquad (32)$$

$$\ddot{\psi} + \frac{\dot{\Phi}}{\Phi}\dot{\psi} = 0. \qquad (33)$$

Consequently, with the use of the constraint equation, we write

$$I(X_2) = \Phi \dot{\psi}, \tag{34}$$

$$\ddot{\Phi} = \frac{(1+V_0)(4+3\omega_{BD})}{3+2\omega_{BD}}\Phi, \tag{35}$$

$$\ddot{a} = (1+V_0)\left(\frac{1+\omega_{BD}}{3+2\omega_{BD}}\right)a + \frac{\dot{a}^2}{a} - \frac{\dot{\Phi}}{\Phi}\dot{a}, \tag{36}$$

or equivalently

$$\dot{H} = (1+V_0)\left(\frac{1+\omega_{BD}}{3+2\omega_{BD}}\right) - \frac{\dot{\Phi}}{\Phi}H. \tag{37}$$

Thus,

$$\dot{H}\Phi + \dot{\Phi}H - (1+V_0)\left(\frac{1+\omega_{BD}}{3+2\omega_{BD}}\right)\Phi = 0, \tag{38}$$

where by replacing from (35), we find

$$(H\Phi)^{\cdot} - \left(\frac{1+\omega_{BD}}{4+3\omega_{BD}}\right)\ddot{\Phi} = 0, \tag{39}$$

that is, the conservation law it follows is

$$(H\Phi) - \left(\frac{1+\omega_{BD}}{4+3\omega_{BD}}\right)\dot{\Phi} = \bar{I}_0 \tag{40}$$

where $\bar{I}_0$ is an integration constant. The latter expression is analoguous to the Noetherian conservation law $I(X_3)$.

Hence, for the scalar field, the analytic solution follows

$$\Phi(t) = \Phi_1 e^{\Omega t} + \Phi_2 e^{-\Omega t}, \ \Omega = \sqrt{\frac{(1+V_0)(4+3\omega_{BD})}{3+2\omega_{BD}}}. \tag{41}$$

For initial conditions, for which $\Phi_1\Phi_2 = 0$, the analytic solution for the Hubble function is

$$H(t) = \mp\frac{\sqrt{1+V_0}(1+\omega_{BD})}{\sqrt{(3+2\omega_{BD})(4+3\omega_{BD})}} + H_0 e^{\pm\Omega t}, \tag{42}$$

respectively. Therefore, the scalar factor is derived to be

$$\ln a(t) = \mp\frac{\sqrt{1+V_0}(1+\omega_{BD})}{\sqrt{(3+2\omega_{BD})(4+3\omega_{BD})}}t \pm \frac{1}{\Omega}e^{\pm\Omega t}. \tag{43}$$

The effective equation of state parameter $w_{eff} = -1 - \frac{2}{3}\frac{\dot{H}}{H^2}$ is calculated.

$$w_{eff}(t) = -1 \mp \frac{2}{3}\frac{\Omega e^{\pm\Omega t}H_0}{\left(\mp\frac{\sqrt{1+V_0}(1+\omega_{BD})}{\sqrt{(3+2\omega_{BD})(4+3\omega_{BD})}} + H_0 e^{\pm\Omega t}\right)^2}. \tag{44}$$

For $H_0 = 0$, it is easy to observe that the de Sitter Universe,

$$\ln a(t) = \mp\frac{\sqrt{1+V_0}(1+\omega_{BD})}{\sqrt{(3+2\omega_{BD})(4+3\omega_{BD})}}t, \tag{45}$$

is recovered. However, for $H_0 \neq 0$ and for large values of $t$, the de Sitter universe is the asymptotic solution.

In general, for $\Phi_1\Phi_2 \neq 0$, the Hubble function is derived to be

$$H(t) = \frac{\sqrt{1+V_0}}{\sqrt{(3+2\omega_{BD})(4+3\omega_{BD})}} \frac{1}{(\Phi_1 e^{2\Omega t} + \Phi_2)} + \frac{H_0}{\Phi_1 e^{2\Omega t} + \Phi_2} \exp(\Omega t). \tag{46}$$

Easily, we observe that the $w_{eff}(t)$ for the latter solution for large values of $t$ asymptotically reaches the value $w_{eff}(t) \to -1$; thus, the de Sitter Universe is an asymptotic solution for the dynamical system.

### 4.2. Model B

For the second model of our analysis, that is, for $(\kappa, \lambda) = (\kappa, 2\kappa)$, $\kappa = \frac{\sqrt{3(3+2\omega_{BD})}}{2} + \frac{3}{4}$, the normal coordinates are $(a, \Xi, \psi)$, in which

$$\Xi = \phi a^{\frac{6}{\sqrt{3(3+2\omega_{BD})}+3}}. \tag{47}$$

In the new variables, the point-like Lagrangian is

$$\mathcal{L}(a, \dot{a}, \Xi, \dot{\Xi}, \psi, \dot{\psi}) = a^{2-\frac{6}{3+\sqrt{3(3+2\omega_{BD})}}} \left( \sqrt{3(3+2\omega_{BD})}\dot{a}\dot{\Xi} - \frac{\omega_{BD}}{2}\frac{a}{\Xi}\dot{\Xi}^2 \right)$$
$$- \frac{1}{2}F_0^2 \Xi^{\frac{3+\sqrt{3(3+2\omega_{BD})}}{2}}\dot{\psi}^2 + V_0 \Xi^{\frac{3+\sqrt{3(3+2\omega_{BD})}}{2}}. \tag{48}$$

Hence, the field equations are

$$a^{2-\frac{6}{3+\sqrt{3(3+2\omega_{BD})}}} \left( \sqrt{3(3+2\omega_{BD})}\dot{a}\dot{\Xi} - \frac{\omega_{BD}}{2}\frac{a}{\Xi}\dot{\Xi}^2 \right) - \frac{1}{2}F_0^2 \Xi^{\frac{3+\sqrt{3(3+2\omega_{BD})}}{2}}\dot{\psi}^2 + V_0 \Xi^{\frac{3+\sqrt{3(3+2\omega_{BD})}}{2}} = 0, \tag{49}$$

$$\ddot{\Xi} = \frac{\sqrt{3(3+2\omega_{BD})} - 3 + \omega_{BD}\left(\sqrt{3(3+2\omega_{BD})} - 2\right)}{2(3+2\omega_{BD})}\frac{\dot{\Xi}^2}{\Xi}, \tag{50}$$

$$\begin{aligned} \ddot{a} &= \frac{V_0\left(\sqrt{3}+\sqrt{3+2\omega_{BD}}\right)}{\sqrt{3+2\omega_{BD}}}a^{-2+\frac{6}{3+\sqrt{3(3+2\omega_{BD})}}}\Xi^{\frac{1+\sqrt{3(3+2\omega_{BD})}}{2}} - \frac{2\sqrt{3(3+2\omega_{BD})}}{3+\sqrt{3(3+2\omega_{BD})}}\frac{\dot{a}^2}{a} \\ &\quad - \frac{27\left(\sqrt{(3+2\omega_{BD})}-\sqrt{3}+\omega_{BD}\left(\sqrt{3}(33+\omega_{BD})+18\sqrt{(3+2\omega_{BD})}\right)\right)}{(3+2\omega_{BD})^{\frac{3}{2}}\left(3+\sqrt{3(3+2\omega_{BD})}\right)}\dot{a}\frac{\dot{\Xi}}{\Xi} \\ &\quad + \frac{\omega_{BD}\left(\sqrt{3}(3+\omega_{BD})+\sqrt{(3+2\omega_{BD})}\right)}{4(3+2\omega_{BD})^{\frac{3}{2}}}a\left(\frac{\dot{\Xi}}{\Xi}\right)^2. \end{aligned} \tag{51}$$

From the second-order differential Equation (50), we derive the conservation law

$$\Xi^{\frac{\sqrt{3(3+2\omega_{BD})}-3+\omega_{BD}\left(\sqrt{3(3+2\omega_{BD})}-2\right)}{2(3+2\omega_{BD})}}\dot{\Xi} = \tilde{I}_1 \tag{52}$$

which is a Noetherian conservation law related to the vector field $\bar{X}_3$, that is, $\tilde{I}_1 \simeq I(\bar{X}_3)$. Thus, for the scalar field $\Xi$, it follows that the closed-form solution is given by

$$\Xi(t) = \Xi_0\left(3 + \sqrt{3(3+2\omega_{BD})} + \omega_{BD}\left(2 + \sqrt{(3+2\omega_{BD})}\right)t\right)^{\frac{2(3+2\omega_{BD})}{3+\sqrt{3(3+2\omega_{BD})}+\omega_{BD}\left(2+\sqrt{(3+2\omega_{BD})}\right)}}. \tag{53}$$

Finally, for the scalar factor, we determine the exact solution

$$a(t) = t^{\frac{5+4\omega_{BD}+\sqrt{3(3+2\omega_{BD})}}{(4+3\omega_{BD})}}. \tag{54}$$

$$V_0 = \frac{2(3 + 2\omega_{BD})}{(4 + 3\omega_{BD})^2} \left( 13 + 9\omega_{BD} + \sqrt{3(3 + 2\omega_{BD})}\Xi_0^{-\frac{1 + \sqrt{3(3 + 2\omega_{BD})}}{2}} \right). \tag{55}$$

Consequently, the Hubble function and the effective equation of state parameter are derived

$$H(t) = \frac{5 + 4\omega_{BD} + \sqrt{3(3 + 2\omega_{BD})}}{(4 + 3\omega_{BD})} \frac{1}{t}, \tag{56}$$

$$w_{eff}(t) = -1 + \frac{22(4 + 3\omega_{BD})}{3\left(5 + 4\omega_{BD} + \sqrt{3(3 + 2\omega_{BD})}\right)}. \tag{57}$$

Therefore, the exact solution describes an accelerated Universe when

$$-\frac{1}{16}\left(\sqrt{33} + 17\right) < \omega_{BD} < \frac{6}{841}\left(\sqrt{22} - 188\right). \tag{58}$$

## 5. Conclusions

The Noether symmetry analysis is a powerful method for the study of nonlinear dynamical systems with a variational principle. The symmetry analysis has been widely applied in gravitational systems for the construction of conservation laws and the study of various cosmological models.

In this study, we applied the Noether symmetry analysis in order to study the nonlinear field equations for a two-scalar field cosmological model in a spatially flat FLRW geometry. The gravitational theory is defined in the Jordan frame, where one of the scalar fields is the Brans–Dicke field, and the second scalar field is minimally coupled to gravity but non-minimally to the Brans–Dicke field. This specific model has been proposed before as the analog in the Jordan frame for the Chiral model, which generates the hyperinflation.

The cosmological model possesses three arbitrary parameters, namely the $(\omega_{BD}, \kappa, \lambda)$. From the application of Noether's theorem, it was found that for specific sets of the variables $(\kappa(\omega_{BD}), \lambda(\omega_{BD}))$, the field equations admit additional conservation laws such that the field equations constitute a super-integrable dynamical system. For these cases, with the use of the normal coordinates, we were able to simplify the field equations and write the closed-form and exact solutions. The analysis of the solutions gives constraints for the free parameter, $\omega_{BD}$, such that the hyperbolic inflationary solution is recovered.

In future work, we plan to investigate further these super-integrable models, and in particular, we plan to solve the Wheeler–DeWitt equation of quantum cosmology and compare the semiclassical limit in the Jordan and Einstein frames.

**Funding:** AP was partially supported by the National Research Foundation of South Africa (Grant Numbers 131604).

**Institutional Review Board Statement:** Not applicable.

**Informed Consent Statement:** Not applicable.

**Data Availability Statement:** Not applicable.

**Conflicts of Interest:** The author declare no conflict of interest.

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
