# Peer review of "Analytic Solution and Noether Symmetries for the Hyperbolic Inflationary Model in the Jordan Frame"

_universe, doi:10.3390/universe8060325_

Round 1

Reviewer 1 Report

I have attached my comments and suggestions as a pdf file.

Author Response

I want to thank the reviewer for the positive comments on my work and for the careful reading.

I have revised my work and the typos mentioned by the reviewer in points 1,2 and 3 have been corrected.

However, I do not agree with the points 4 and 5 of the reviewer. It order to present in a simple form the conservation laws related by Noether symmetries, the new expressions (40) and (52) are the corresponding Noetherian conservation laws which are used for the reduction of the dynamical system. I believe that now the analysis is clear for the reviewer and the reader and that my work is suitable now for publication. 

Reviewer 2 Report

The authors recall the Noether symmetry analysis to investigate the nonlinear field equations for a two-scalar field cosmological model in a spatially flat FLRW geometry. The gravitational theory is defined in the Jordan frame, where one of the scalar fields is the Brans-Dicke field, and the second scalar field is minimally coupled to gravity but nonminimally to the Brans-Dicke field. 
The examination of the solutions leads to constraints for the free parameter, and the hyperbolic inflationary scenario is recovered.

The paper is well motived and the Noether formalism is pertinent, the results are clearly presented. 

Author Response

I want to thank the reviewer for the positive comments on my work. The manuscript has gone under a native English speaker and minor typos have been corrected. 

I believe that this version of the manuscript is suitable for publication. 

Round 2

Reviewer 1 Report

I have attached my comments and suggestions for author as a pdf file. 

Author Response

I want to thank the reviewer for the additional comments on my work. During the previous revision, I constructed the conservation laws in the new variables. 

I this reply, I attach a pdf file, which is an export from Maple, where I verify the Noether symmetries for my analysis. The additional symmetries that I have found in my work are with \xi=0, f=0 and eta_t=0. Hence, we can make use the results of arXiv:1101.5771 and easily verify that the specific functional forms that I found before admit additional Noether symmetries.

I believe that now I have clarified the main concerns of the reviewer.

p.s. There are some typo in the Noether symmetry conditions given by the reviewer.

Round 3

Reviewer 1 Report

I have enugh clarification for my concerns. So, I would like to recommend  this article to be published in the journal Universe.